# An international consensus on effective, inclusive, and career-spanning short-format training in the life sciences and beyond

Jason J. Williams[1]☯*, Rochelle E. Tractenberg[2]☯*, Bérénice Batut[3,4], Erin A. Becker[5], Anne M. Brown[6], Melissa L. Burke[7,8,9], Ben Busby[10], Nisha K. Cooch[11], Allissa A. Dillman[12], Samuel S. Donovan[13], Maria A. Doyle[14,15], Celia W. G. van Gelder[16], Christina R. Hall[7,17], Kate L. Hertweck[18], Kari L. Jordan[5], John R. Jungck[19], Ainsley R. Latour[20], Jessica M. Lindvall[21,22], Marta Lloret-Llinares[23], Gary S. McDowell[24,25,26], Rana Morris[27], Teresa Mourad[28], Amy Nisselle[29,30], Patricia Ordóñez[31], Lisanna Paladin[32], Patricia M. Palagi[33], Mahadeo A. Sukhai[34,35], Tracy K. Teal[36], Louise Woodley[37]

1 DNA Learning Center, Cold Spring Harbor Laboratory, Cold Spring Harbor, New York, United States of America, 2 Collaborative for Research on Outcomes and Metrics, Georgetown University, Washington, DC, United States of America, 3 Albert-Ludwigs-University Freiburg, Freiburg, Germany, 4 Open Life Science, Freiburg, Germany, 5 The Carpentries, 6 Virginia Tech, Blacksburg, Virginia, United States of America, 7 Australian BioCommons, North Melbourne, Australia, 8 Queensland Cyber Infrastructure Foundation, Research Computing Centre, 9 The University of Queensland, 10 DNAnexus, Mountain View, California, United States of America, 11 CureComms Advisors, LLC, 12 BioData Sage, 13 BioQUEST Curriculum Consortium, 14 Bioconductor, 15 University of Limerick, Limerick, Ireland, 16 Dutch Techcentre for Life Sciences, Utrecht, The Netherlands, 17 University of Melbourne, Melbourne, Australia, 18 Chan Zuckerberg Initiative, Redwood City, California, United States of America, 19 University of Delaware, Newark, DE, United States of America, 20 IDEA-STEM, 21 ELIXIR-SE/NBIS, 22 Science for Life Laboratory Training Hub, Solna, Sweden, 23 European Molecular Biology Laboratory, European Bioinformatics Institute, Cambridge, United Kingdom, 24 Lightoller LLC, 25 The Ronin Institute, Montclair, NJ, United States of America, 26 Institute for Globally Distributed Open Research and Education, 27 National Center for Biotechnology Information, National Library of Medicine, National Institutes of Health, 28 Ecological Society of America, Washington, DC, United States of America, 29 Murdoch Children's Research Institute, Melbourne, Australia, 30 Melbourne Genomics, The University of Melbourne, Melbourne, Australia, 31 University of Maryland Baltimore County, Catonsville, Maryland, United States of America, 32 European Molecular Biology Laboratory, Structural and Computational Biology Unit, Heidelberg, Germany, 33 Swiss Institute of Bioinformatics, Lausanne, Switzerland, 34 Canadian National Institute for the Blind, Toronto, Canada, 35 Queen's University School of Medicine, Kingston, Canada, 36 Posit, PBC, Boston, Massachusetts, United States of America, 37 Center for Scientific Collaboration and Community Engagement, Oakland, California, United States of America

☯ These authors contributed equally to this work.
* williams@cshl.edu (JJW); rochelle.tractenberg@georgetown.edu (RET)

**Data Availability Statement:** All relevant data are within the paper and its Supporting information files. Per Delphi best practices (now cited in the

## Abstract

Science, technology, engineering, mathematics, and medicine (STEMM) fields change rapidly and are increasingly interdisciplinary. Commonly, STEMM practitioners use short-format training (SFT) such as workshops and short courses for upskilling and reskilling, but unaddressed challenges limit SFT's effectiveness and inclusiveness. Education researchers, students in SFT courses, and organizations have called for research and strategies that can strengthen SFT in terms of effectiveness, inclusiveness, and accessibility across multiple dimensions. This paper describes the project that resulted in a consensus set of 14 actionable recommendations to systematically strengthen SFT. A diverse international

paper), intermediate outputs are not considered shareable data, so those are not available.

**Funding:** This work was supported by the National Science Foundation under DRL/EHR: 2027025 to JWW & RET. This funder played no role in the study design, data collection and analysis, decision to publish, or preparation of the manuscript. Any opinions, findings, and conclusions or recommendations expressed in this material are those of the author(s) and do not necessarily reflect the views of the National Science Foundation. Several co-authors are employed by commercial entities. These employers provided support in the form of salaries for BB, KLJ, EAB, and NC and did not have any additional role in the study design, data collection and analysis, decision to publish, or preparation of the manuscript. The specific roles of these authors are articulated in the 'author contributions' section.

**Competing interests:** This work was supported by the National Science Foundation under DRL/EHR: 2027025 to JWW & RET. This funder played no role in the study design, data collection and analysis, decision to publish, or preparation of the manuscript. Any opinions, findings, and conclusions or recommendations expressed in this material are those of the author(s) and do not necessarily reflect the views of the National Science Foundation. Several co-authors are employed by commercial entities. These employers provided support in the form of salaries for BB, KLJ, EAB, and NC and did not have any additional role in the study design, data collection and analysis, decision to publish, or preparation of the manuscript. The specific roles of these authors are articulated in the 'author contributions' section.

group of 30 experts in education, accessibility, and life sciences came together from 10 countries to develop recommendations that can help strengthen SFT globally. Participants, including representation from some of the largest life science training programs globally, assembled findings in the educational sciences and encompassed the experiences of several of the largest life science SFT programs. The 14 recommendations were derived through a Delphi method, where consensus was achieved in real time as the group completed a series of meetings and tasks designed to elicit specific recommendations. Recommendations cover the breadth of SFT contexts and stakeholder groups and include actions for instructors (e.g., make equity and inclusion an ethical obligation), programs (e.g., centralize infrastructure for assessment and evaluation), as well as organizations and funders (e.g., professionalize training SFT instructors; deploy SFT to counter inequity). Recommendations are aligned with a purpose-built framework—"The Bicycle Principles"—that prioritizes evidenced-based teaching, inclusiveness, and equity, as well as the ability to scale, share, and sustain SFT. We also describe how the Bicycle Principles and recommendations are consistent with educational change theories and can overcome systemic barriers to delivering consistently effective, inclusive, and career-spanning SFT.

## Introduction

A shared characteristic of science, technology, engineering, math, and medical (STEMM) disciplines is that "new technologies replace the skills and tasks originally learned by older graduates" and "technological progress erodes the value of these skills over time [1]."

For example, advanced computational methods such as machine learning have transformed life science with 1,487 publications on PubMed referencing this technique in 2012, compared to 30,684 in 2022 [2]. This level of disruptive change can leave practitioners at risk of having large areas of their discipline rendered unintelligible to them [3–5]. Life scientists see computational and data management training as their most unmet need [6, 7], reflecting the challenge in modern science to incorporate knowledge and skills from across multiple disciplines (e.g., computational methods, see [8].

This project explored the application of evidence-based teaching and principles of inclusion and equity to improve short-format training (SFT) such as workshops, bootcamps, and short courses (full definition in Supplemental Information, S1 Text). SFT is widely used for upskilling and reskilling in rapidly evolving disciplines such as life science where disruptive changes and shifting skill sets are increasingly common. SFT's popularity can be attributed to several positive features such as its relatively low cost and time commitment, as well as its capacity for rapid update and customization. Given the urgent need for full participation in STEMM (e.g., [9–11]), it is also important to note that SFT can be designed or revised to equitably include historically excluded people. For example, *The Carpentries Toolkit of IDEAS* provides strategies before, during, and after SFT to promote inclusion, diversity, equity, and accessibility [12]. In addition to purely technical skills, SFT is also used to disseminate and reinforce professional practices such as research rigor, reproducibility, and other open science skills [13, 14]. Common abbreviations and definitions for this project and manuscript appear in Table 1.

Despite its positive features, SFT's efficacy—its ability to measurably improve learners' knowledge, skills, and abilities—may be much lower than is commonly realized. Feldon et al. [15] is the most extensive independent and peer-reviewed study to date that systematically

**Table 1. Abbreviations and definitions.**

| |
| --- |
| • **STEMM**: Science, Technology, Engineering, Math, Medicine* |
| • **SFT**: Short-format training; SFT involves instruction in disciplinary skills and knowledge over a relatively short duration (i.e., hours, days, or a few weeks). Rather than specifying a set number of hours, the easiest way to identify SFT is that it will be labeled as a workshop, bootcamp, short-course, or similar term. We generally do not include short vocational training or continuing medical education, which have regulated formal requirements. See an expanded definition in Supplemental Information (S1 Text). |
| • **FHE**: Formal higher education; formal education associated with undergraduate or graduate degrees. |
| * **Note** Where direct quotes are used, the abbreviation STEM may appear, which does not explicitly exclude medicine. |

evaluated the impact of SFT on life scientists. This study analyzed SFT interventions involving 294 life science Ph.D. students from 53 U.S. institutions across 115 variables and found "no evidence of effectiveness." Feldon et al. concludes that "boot camps and other short formats may not durably impact student outcomes," and that more effort and resources should be spent on improving SFT.

Feldon's findings align with prior research in and beyond the U.S. (e.g., [16, 17]). The 2022 *5th Global Report on Adult Learning and Education* of the UNESCO Institute for Lifelong Learning [18] notes that only 60% of participating EU countries use learning outcomes as a quality measure of adult learning and education, across all types of instruction. Quality assessment is recognized to be difficult "...because of the diversity and plurality, and sometimes decentralized and deregulated nature, of the field—not to mention the variety of learners' aims —across national and regional settings." ([18] p. 25). We do not assert that all SFT is ineffective. However, we know from other STEMM instructional settings that some learners are likely to benefit from a learning opportunity no matter how well or how badly it is taught. Cooper et al. [19] notes that, "(a)lthough most STEM faculty and practicing scientists have learned successfully in a traditional format, they are the exception, not the norm, in their success" (p. 281). If instruction is only "effective" for learners who are unaffected by the quality of instruction, then that instruction is literally exclusionary because not all learners will benefit.

There is a strong rationale for reforming SFT. SFT's positive features satisfy needs that are difficult or impossible to address otherwise. There is consistent demand for SFT training opportunities worldwide (e.g., [7]), and university, research institutes and government agencies continue to provide substantial funding for SFT; from 2017–2022 GrantExplorer reported expenditure of $4 billion in NSF, $83 million in NIH, and $767 million in DoD funding to projects associated with some SFT output [20–22].

Currently, STEMM education reform focuses primarily on formal higher education (FHE) [23–27], but these efforts are unlikely to directly impact SFT. Contrasting FHE and SFT (see Fig 1) and noting the variabilities identified by UNESCO [18] suggests that techniques used to improve FHE, if feasible for SFT, would likely require modification. SFT is not simply a "short" version of instruction in FHE; their only shared characteristic is that formal knowledge about teaching and learning ought to apply to both (e.g., [28, 29]). Considering the features of FHE holistically, it should also be noted that FHE's relative uniformity makes it easier to develop systemic reforms; the ability to address systemic problems is an additional reason FHE is more concretely improvable than SFT. As Reinholz et al. [30] suggests, "The goal of improving postsecondary STEM education requires careful attention to many interlocking systems and parts of systems." (see also [31]; p. 952; [32, 33]). Efforts to improve STEMM instruction in FHE have proceeded with some assurance that findings and interventions could be generalized across similar institutions and programs. This generality is more difficult for

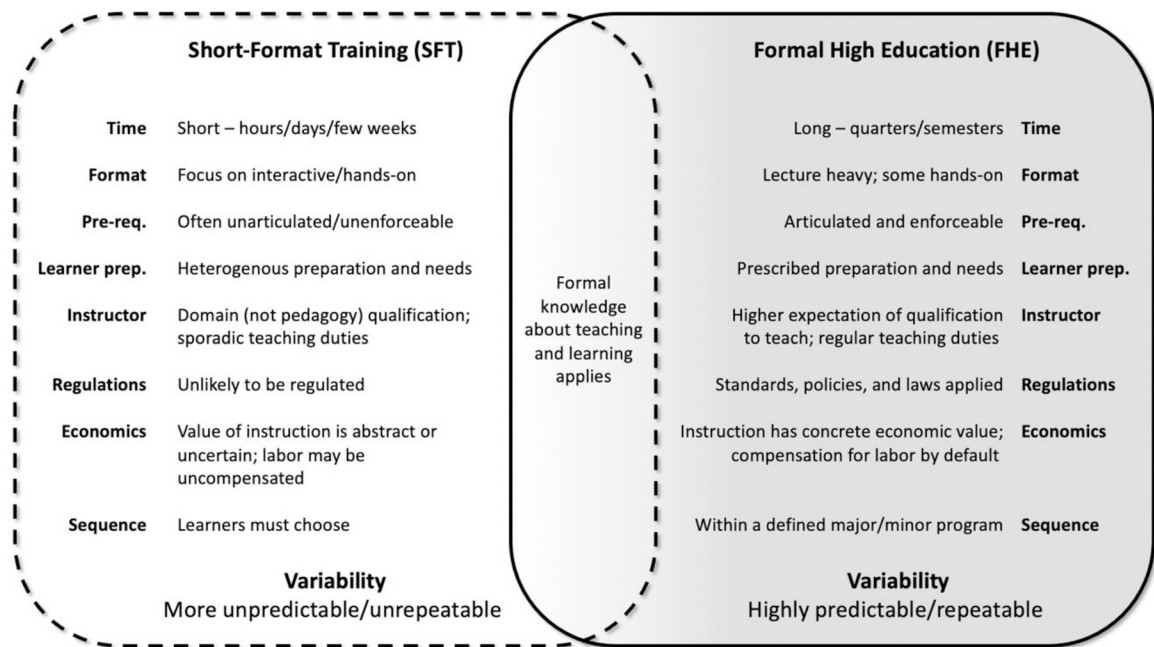

**Fig 1. Differences and similarities between short-format training (SFT) and formal higher education (FHE).** SFT is highly variable: contrasting features of short-format training (SFT) and formal higher education (FHE). We wrap SFT with a dashed-line highlighting that any given SFT may be difficult to define exactly (see expanded definition of SFT, Supplemental Information, S1 Text); FHE's solid line indicates its greater uniformity and lower variation. The **Time** of instruction in SFT is short—from hours to a few weeks—vs. many weeks for a course in FHE. The **Format** of SFT in the life sciences is typically focused on some form of active learning vs. more traditionally lecture-heavy FHE. **Prerequisites** are easier to define and enforce in FHE, unlike in SFT. **Learner preparation** is also difficult to predict in SFT, vs. FHE where learners will have been predictably prepared by prior coursework or the course is designed to be foundational. **Instructors** in SFT are often domain experts but may have limited teaching experience; FHE instructors will usually have some expectation of preparation for teaching, may have been teaching the material regularly, and often have the benefit of access to, or being expected to use, expert assistance in the planning and delivery of instruction. **Regulations**, policies, and laws usually apply to FHE courses; frequently, the informal nature of SFT is not affected by these and in practice SFT may not be regulated the same way as formal classroom instruction. **Sequence** of classes in a FHE curricular program provides learners with clear guidance on next steps, whereas SFT learners must direct their own learning; there may be additional SFT opportunities that can assist them in achieving their objectives, but this is not common. **Economics** of FHE assigns a concrete value to instruction; features of instruction, such as quality, can command a higher price; incentivizing maximized quality. Instructional effort is generally compensated. SFT instruction may be made available without cost to learners, but this may result in an underestimation of the value or quality of instruction; there may be no economic incentives for optimizing "free" instruction. Instruction delivery may rely on uncompensated volunteer labor. **Variability** is the summation of all these characteristics. The variability of SFT is high [18], two courses on a similar topic may differ widely; instructional practices and curricula may not be documented for reuse. In FHE, variability in these characteristics is much lower; comparisons can be made across programs, allowing equivalencies and credit exchanges across institutions and programs. It is possible to make formal comparisons between distinct FHE programs (e.g., [34]) within one country or university system. Transferring from one higher education institution to another involves a systematic assessment of equivalence of prior work (e.g., [35]; see also [36]). This figure emphasizes the fact that perhaps the only shared characteristic of these instructional forms is that formal knowledge about teaching and learning ought to apply to both; and that strategies for improving FHE STEMM education would likely not be transferable to SFT without significant modification.

SFT where FHE's "systems and parts of systems" may be difficult to compare, unrecognizable, or non-existent.

Compared to FHE, the variability of SFT makes it far more difficult to address as a system. As noted by UNESCO [18], this variability arises on both the instruction side ("diversity, plurality,. . . decentralized and deregulated nature") as well as from the learner side (*viz.* "variety of learners' aims"). Except for a few large-scale SFT programs with an explicit focus on instructional quality, SFT instructors may have little knowledge or understanding of learner preparedness and contexts. SFT instructors are generally chosen for domain expertise and

may not have pedagogical training or support that could help them adapt their teaching to overcome obstacles they encounter. Additionally, SFT courses are often bespoke, independent, and transient. This increases the chance that even effective instructional content and practices are unimplemented, unshared, and difficult to replicate. For learners, it may be impossible to compare different SFT opportunities on the same topic, meaning that decisions to enroll are based primarily on what is available. Learners wishing to prioritize effectiveness, accessibility, and inclusivity of instruction may also lack information or assurances on these characteristics in advance of the training. Overall, SFT lacks the stabilizing pedagogical, programmatic, policy, and economic structures that make improvement in the FHE context more tractable. Any given SFT course is therefore at risk of being a "black box," having a definite form (i.e., short) but unknown contents (e.g., effectiveness and inclusiveness).

Since SFT lacks the system-context that is crucial to FHE reform (e.g., Reinholz et al. [30]), SFT reform could benefit from an approach that systematizes SFT. Rather than imposing FHE structures on a vastly different instructional context, systematization could be achieved by identifying features SFT programs have in common and designing interventions that address problems from multiple angles. For SFT, it is reasonable to conclude that reform efforts should engage the entire set of stakeholders (e.g., learners, instructors, instruction designers, administrators, funders) that make up the SFT "system." Reforms that are actionable for both individuals and collectives have more possibilities for implementation. Recommended changes could be designed as standalone measures (e.g., a change an individual instructor could implement), or achieve impact as groups of people adopt them (e.g., shared sets of standards or credentials).

Optimizing SFT for effectiveness and inclusion across the career span is timely and justified. The U.S. National Science Foundation 2026 Idea Machine project (NSF 2026) identified "high impact grand challenges" in research and STEM education that could help "set the U.S. agenda for fundamental research [37]." The research presented here emerged from the "Reinventing Scientific Talent" proposal, which was selected as an NSF 2026 grand challenge. This proposal called for the "transform[ation] of the education of scientists and STEM professionals after their formal training." A small think tank-style conference was designed to assemble representative global efforts in SFT to generate actionable recommendations for improvement.

## Materials and methods

The study was approved (exempted) by the Georgetown University Institutional Review Board (IRB# STUDY00003859); To structure conference discussions, project PIs (J.J.W., R.E.T.) synthesized a draft set of principles from literature and experience which were further refined by the Organizing Committee (Organizers: J.J.W., R.E.T., and B.B., S.S.D., K.J.L., T.M., T.K.T., C. vG.). The Bicycle Principles synthesize education science and community experience into a framework for improving SFT through two cyclic (hence "bi-cycle") and iterative processes (see Fig 2).

### Recruitment

Participants were recruited through a widely advertised self-nomination process and by direct invitation. The announcement was distributed to colleagues, communities of interest, and through social media (see Supplemental Information, S2 Text). Nominations were accepted April 14th through May 31st, 2021, using a form also completed by direct invitees. Through literature search, PIs identified and contacted 31 additional candidates with relevant expertise. Excluding conflicts of interest, organizers scored and ranked applicants. Participants who increased non-overlapping areas of expertise and added to gender, ethnic, and racial diversity

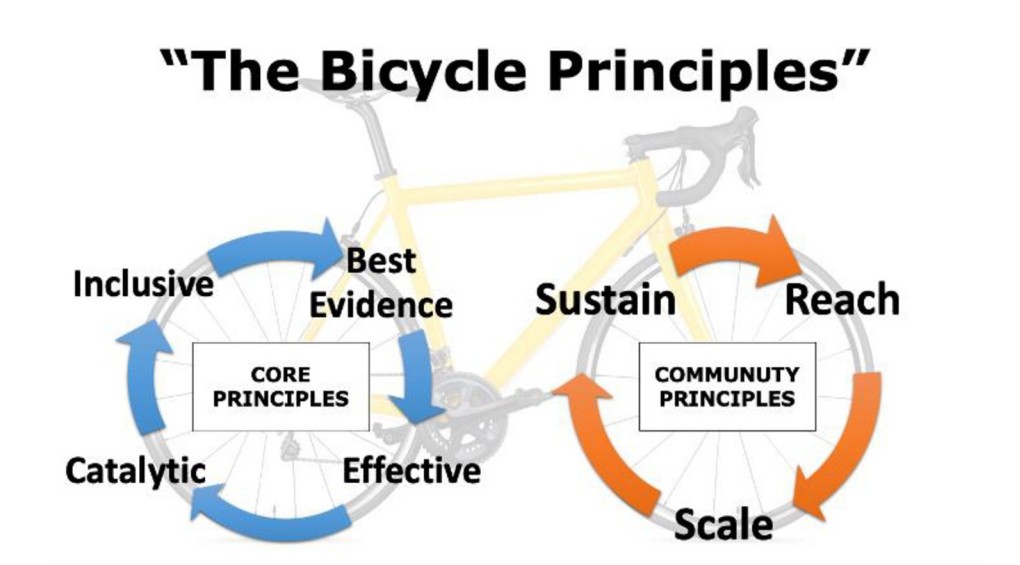

**Core Principles** apply to all short-format training:

**All short-format training should...**

- Use <u>Best Evidence</u>; ground instruction in findings from the education sciences and formally evaluated instruction.
- Be <u>Effective</u>; provide evidence (i.e., from assessment, evaluation) to learners that they have made progress in achieving programmatic and learning goals.
- Be <u>Inclusive</u>; maximize the ability of all learners to participate in and benefit from the learning experience.
- Promote <u>Catalytic</u> learning; prepare learners to succeed when the application of knowledge, skills, and abilities requires further self-directed study.

**Community Principles** apply when SFT is developed and/or deployed by communities or other groups to achieve specific objectives:

**Communities increase the impact of SFT by working to...**

- <u>Reach</u>; include new types and larger audiences of learners.
- <u>Scale</u>; increase delivery of short-format training by new groups and larger numbers of instructors and instructional developers.
- <u>Sustain</u>; work to maintain the availability, usability, relevance, and reliability of learning materials as well as supporting the supporting infrastructures, trainers, and communities which enable effective and inclusive training.

**Fig 2. The Bicycle Principles.** "The Principles" represent a framework for recommendations to improve SFT. One cycle comprises "Core Principles" that all SFT should meet, that is, SFT should be: (based on) *Best Evidence*, *Effective*, *Inclusive*, *Catalytic*. The second cycle of "Community Principles" apply when the SFT is developed with the potential or intention to be reused and disseminated beyond the original or initial deployment, that is, it should promote: *Reach*, *Scale*, and *Sustain*. These two iterative cycles make up the "bicycle".

were prioritized. A small number of participants (virtual and in-person) from policymaking and funding agencies or bodies were also recruited. Participants also included representatives from large scientific professional societies as well as private sector companies. In areas lacking representation (e.g., ethnicity, experience), additional invitations were sent, and selection concluded by September 2021. Our budget supported 20 in-person participants and virtual participants up to the intended cap of 30–35 total (to encourage full participation in discussion). Notably, recruitment, nomination, and selection process occurred during the COVID-19 pandemic limiting potential participants.

## Meeting 1 (100% virtual)

Although we originally planned a single meeting, this virtual kick-off meeting took advantage of the postponement of the in-person conference due to COVID-19. The organizers generated 20 vignettes (i.e., brief statements of training-related situations) see Supplemental Information, S3 Text) on challenges associated with SFT based on content analysis with phenomenography of the vignettes compiled from nomination forms, together with others synthesized from the experiences of the PIs (J.J.W., R.E.T.) and discussions with the organizing committee. Phenomenography is a qualitative research technique applied to better understand the variety with which individuals experience or understand a common construct or phenomenon [38]. Prior to the kick-off, participants provided feedback on the accessibility of the virtual meeting tools. Work was captured in virtual whiteboards and Slack chat. Participants also received a *precis* (see Supplemental Information, S4 Text), the vignettes (S2 Text), literature underpinning The Principles (including [39, 40]), and summarized conference goals. One Principles-supporting white paper described the construct of *catalytic learning*, defined as learning that, once completed, enables the learner to continue learning in a self-directed way [40]. To accommodate most time zones, two sessions were held via Zoom in December 2021. During the kick-off participants selected vignette(s) that they felt were aligned with their interests and expertise. Next, participants broke out into virtual rooms to discuss, develop, and justify recommendations they felt could enhance SFT's effectiveness, inclusiveness, relevance across the career span, or some combination of these features. The PIs examined all kick-off meeting outputs and applied content analysis to discern emergent themes. Outputs were examined by the PIs (J.J.W., R.E.T) independently, utilizing the scripts and activity outlines that were developed to guide the virtual meetings to deduce themes emerging from our informal quantitative content analysis [41] p. 233. While our content analysis was informal, it was designed and executed independently to generate a "careful, detailed, systematic examination and interpretation of a particular body of material in an effort to identify patterns, themes, biases, and meanings" [42] p.349. Themes from the virtual meetings that both PIs identified were retained. The results of these kick-off meeting outputs were synthesized into a set of 19 recommendations to work from at the main meeting.

## Meeting 2 (70% in person, 30% virtual)

In May 2022, a three-day hybrid conference (in person plus virtual attendance) was held at the Cold Spring Harbor Laboratory Banbury Center in New York. Over the three days, participants were invited to give 14 presentations on how The Principles and specific recommendations had been or might be implemented within their various represented programs or presented feedback on The Principles in the context of their professional areas of expertise. The 19 recommendations that had been drafted based on the kickoff were used as a starting point for participants to develop and refine recommendations. All participants attended the

presentations, and were able to access, and comment on, the vignettes and recommendations as they evolved during the meeting.

The conference included an active, two-phase, "live" Delphi. A Delphi method is a systematic and qualitative approach to eliciting expert opinions through structured questions and interactions, so ideally suited to this project. In the first phase, each recommendation was assessed in real time by participants. Judgements were elicited about the recommendations under consideration from within a structured Google Doc where each recommendation had an identical template, like a worksheet with the vignette at the top. We asked for both input and evidence supporting the input, on each vignette. More than one participant was working on a recommendation to address each vignette at once, actively coming to consensus on definitions, wording, references, and responses to our requests for evidence that a recommendation would be feasible and have the intended impact(s). Evidence could reflect experience of participants with other, similar policies or structures, or could be support from peer reviewed or grey literature (e.g., books, chapters, white papers). Additionally, we asked for details on what "success" might look like if the recommendation were implemented, as well as barriers, incentives, and other considerations for each recommendation across multiple stakeholder groups. As the vignettes and recommendation worksheets were filled in, multiple contributors offered an authentic "member check in" on each other's work. In this way, Delphi input was actively evolving towards consensus as it was entered; individuals answered the series of prompts and were able to (and often did) go back to refine earlier statements. We encouraged participants to collaborate on the recommendations they were most interested in or had the most experience with (or both); but throughout the meeting each recommendation was evaluated by every participant (dynamic member check-in). The second phase of the Delphi was driven by a more formal, quantitative, content analysis by one author (R.E.T.), with ongoing member check-in with any participants who had contributed to a recommendation.

Intermediate conference outputs included notes compiled by participants and a dedicated science writer (N.C.) (15,488 words, plus additional comments); three participant-synthesized papers on Catalytic learning, Inclusion, and Scaling/Sustaining training (A.L., G.S.M., L.P., M. S., S.D., and J.J.W., R.E.T; 13,760 words, plus additional comments). The activities of the conference generated a recommendation synthesis document (10,944 words, plus additional comments). Extensive documentation and participation by all helped ensure saturation with respect to recommendations on each vignette and established that recommendations were sufficiently detailed for actionability across diverse contexts (e.g., academic, federal, business settings, and globally). Consistent with Delphi best practices, we encouraged full and free engagement with this task by ensuring we would not share these intermediate outputs publicly [41 p. 113, 43 p. 126].

## Results

The two-stage conference assembled recognized experts in SFT, FHE, and the cognitive and educational psychology of higher education. Participants included several of the largest SFT programs reaching life scientists today (e.g., *The Carpentries* [44], *ELIXIR* training [45], *Galaxy Training Network* [46], *Cold Spring Harbor Laboratory* and *DNA Learning Center* courses [47, 48]), as well as emerging programs in SFT training, related research, and funding. Participants' experience included SFT program development, deployment, and revision, as well as expertise in disability, equity, and inclusion. Some participants are engaged within their home countries, but most work overseas training in international settings. The content areas in which the participants' training efforts are focused include bioinformatics, computational biology, computer science, data analysis, data science, genetics, molecular biology, and STEMM education in

undergraduate and/or graduate contexts. There were 30 participants from ten countries: U.S. and Puerto Rico, Australia, Canada, France, Germany, The Netherlands, New Zealand, Sweden, Switzerland, and the United Kingdom, with similar participation for the in-person conference. Participant-reported demographic information included Gender: 73% Women, 23% Men, 3% Non-binary; Race/ethnicity: 73% White, 10% Asian, 10% Black, 3% Hispanic, 3% Indigenous; Other categories: 23% Underrepresented in the sciences, 7% disabled, 7% other identities. Post-conference, participants detailed the reach of their respective SFT programs. Participants' programs train between 0 and 7000 new trainers, and train or reach between 0 and 8000 learners in any given year. Among participants' organizations, *The Carpentries* SFT program reaches the largest number of countries: 64 [49], with most having more limited geographic coverage (range of countries in which participants' groups operate: 1–64).

At the virtual kickoff meeting, three themes emerged as most important for the participants, derived from the 20 initial vignettes. The participants in both virtual kick-off sessions quickly converged on the same three areas:

1. **Catalytic Learning**: How SFT can better position learners to be self-directed after the completion of a training event [40].

2. **Inclusion**: How SFT can be made more inclusive for learners of diverse backgrounds and abilities.

3. **Scaling and Sustaining**: How, and with the help of what incentives, effective SFT can be sustainably scaled to large numbers of learners by large numbers of instructors.

The Bicycle Principles (Fig 2) were developed by the project principal investigators with discussion among the organizing committee. Through discussion and minor modifications for clarity, The Bicycle Principles were adopted as the guiding framework for drafting recommendations that addressed the three key themes from the kickoff meeting.

At the hybrid conference, participants used a Delphi procedure to develop recommendations, in alignment with The Bicycle Principles, for improving the efficacy and inclusivity of SFT across the career span through attention to each of these three areas. From the 19 draft recommendations at the start of 2nd meeting, this group derived a final list of 14 actionable recommendations (Tables 2–5) for systematic, evidence-based improvements to SFT that are aligned with the Bicycle Principles framework.

The group collaboratively generated a document outlining 14 recommendations, where each recommendation is elaborated through six descriptive sections: 1) *Summary* expands upon the problem the recommendation tries to solve; 2) *How might this work* presents an implementation example and suggestions on evaluating success; 3) *Related Principles* lists closely related Bicycle Principles; 4) *Benefits to the learners* lists how recommendation helps learners (directly or indirectly); 5) *Incentives to Implementers* lists motivations for implementers to enact this recommendation; and 6) *Barriers to Implementation* lists obstacles that may

**Table 2. Recommendation starting points: Individual SFT instructors—Grass roots.**

| Recommendation | Title |
| --- | --- |
| F | Make the Bicycle Principles actionable for funders |
| I | Apply FAIR Principles* to training materials |
| M | Support integration of diagnostic assessment into short-format training |
| L | Develop an implementation strategy for Catalytic learning |

*Findable, Accessible, Interoperable, Reproducible (FAIR) Principles [51]

**Table 3. Recommendation starting points: Group/community—leading from the middle.**

| Recommendation | Title |
|---|---|
| A | Professionalize the training of short-format training instructors and instructional designers |
| G | Clarify the economic models that enable short-format training |
| H | Document models for high-fidelity reaching, scaling, and/or sustaining short-format training |
| K | Communicate standards of instruction through badging |

**Table 4. Recommendation starting points: Organizations and institutions—Top-down.**

| Recommendation | Title |
|---|---|
| B | Centralize infrastructure for short-format training assessment and evaluation |
| C | Support microcredentialing of short-format training instructors |
| J | Encourage interoperable short-format training registries |
| N | Encourage evidence-based guidance to support career-spanning learning |

**Table 5. Recommendation starting points—Action at all levels.**

| Recommendation | Title |
|---|---|
| D | Operationalize equitable and inclusive practice in short-format training as an ethical obligation |
| E | Deploy short-format training to counter inequity |

hinder implementation of this recommendation. We provide a full example for Recommendation (A) in Supplemental Information (S5 Text) and the entire set of recommendation descriptions is available at bikeprinciples.org [50]. The 14 recommendations do not have an intrinsic ordering, and when considering which stakeholder types might have the greatest likelihood of success with implementation of any one or more of the recommendations, groupings emerged. These groupings, and suggestions for implementation for each recommendation by the different stakeholder types, are elaborated in a Roadmap document [46]. The next four tables (Tables 2–5) summarize grouping of the recommendations according to this likelihood of success by different implementers, that is, instructors at the grass roots; professional groups and communities of practice, leading from the middle; or formal organizations and institutions (defined below), top-down. All 14 recommendations appear in Tables 2–5, and the full list (ungrouped) appears in the Supplemental Information (S6 Text).

While individual instructors can implement any of the 14 recommendations, recommendations I, M, and L (Table 2) may be most successful when implemented by individual instructors. Data from an instructor's own courses or scholarship could help deliver empirical results that justify (future or broader) implementation of recommendations by larger groups or institutions. These data are likely needed to convince funders, so recommendation F may also be most successful when the first implementation is by individual instructors or instructional developers.

Groups or communities of practice that are loosely structured (e.g., comprised mostly of volunteers) may have the highest likelihood of success implementing four other recommendations, listed in Table 3.

Community or group efforts (i.e., groups with primarily volunteer, rather than contractual, arrangements with members) may be most supportive of recommendations A, G, H, K. One reason is that, without community-level endorsement, individual instructors might find it

difficult to rally other instructors around shared goals and standards. Moreover, groups and communities currently engaged in professionalizing the training SFT instructors (e.g., *The Carpentries*, *ELIXIR*) might be best positioned to achieve the broad buy-in needed to create and communicate standards of instruction through badging as well as describe economic models that underpin their SFT programs. Leadership "from the middle" on these four recommendations in particular can support individual implementers and can also facilitate adoption of these recommendations by formal organizations and institutions.

Organizations and institutions (e.g., employers or groups with formal contracts or understanding with individuals, groups/communities, or other organizations) could certainly implement the recommendations shown in Table 3 (i.e., A, G, H, K). With their formal structure and infrastructure, organizations and institutions are perhaps best positioned to implement four other recommendations shown in Table 4.

Organizations and institutions would be able to implement complex recommendations which require teams of experts and sustained funding. These same groups can also advance other recommendations, either promoting and supporting grassroots efforts, or encouraging wider adoption once communities and/or individuals have laid sufficient groundwork.

The remaining recommendations (Table 5) are actionable at all levels.

There are fruitful actions to be taken at all levels to implement these two recommendations. Each stakeholder group would be equally likely to be successful with implementing recommendations D and E.

## Discussion

This work addresses unanswered calls to improve SFT efficacy [9, 10, 15, 18] as well as the pressing need for educational reforms to improve equity and inclusion [11, 18, 52]. We also underscore the increasing need to extend the attention of STEMM reform beyond undergraduate and graduate education. The NSF 2022–2026 strategic plan [53] calls for "research that will develop and test new models for the lifetime integration of career and technical training, to keep pace with the ever-expanding frontiers of knowledge" and states that research on "how learning can continue throughout a person's lifetime is crucial if we are to exploit these opportunities and maintain a competitive economy (p. 16)." The Bicycle Principles and 14 recommendations can support such research as they present testable assertions with evaluable impacts on learners.

Cognizant of the difficulties FHE STEMM reform has faced, and that SFT's variability makes it even more complex to improve, there are two questions we should consider. First, why might this effort succeed in improving SFT in general? Second, what can be done to increase the likelihood that the Bicycle Principles and the recommendations are used?

The Bicycle Principles and recommendations can succeed because they serve as tools for making SFT measurable and standardizable. Requiring metrics and standards are a necessary step for any reform, and establishing a common set of principles creates reference points without insisting on rigid inflexibility. Without evaluable definitions for effective and inclusive SFT, it would be impossible to determine if and to what extent any change effort is successful. To quote an aphorism, "If you can't measure it, you can't improve it."

The core Bicycle Principles demand that SFT is grounded in evidence-based teaching (*Best evidence*) and measured by evaluation and assessment (*Effective*). These Principles are explicit in many recommendations (e.g., B, C, H, K, M) and are consistent with curriculum and instructional guidelines [39].

Furthermore, The Bicycle Principles require instructors to consider if SFT is an appropriate format for instruction. SFT should not be used—at least as the sole mechanism for instruction

—when it is not compatible with the intended learning outcome(s). Incremental updates or "just-in-time" training is very compatible with SFT. Learners seeking complex sets of skills or retraining for proficiency in a new discipline have more complex needs and may need more than a "short" amount of time. The *Catalytic* principle and recommendation (L): *Develop an Implementation Strategy for Catalytic Learning*, encourage instructors to work to support self-directed learning beyond the end of the learning experience. For SFT this is essential since, by its nature and time limitations, desired learning outcomes will often exceed what SFT can deliver on its own. Recommendations (H, I, J, M) would (also) support learners in identifying additional SFT and other learning materials that could help them after an introductory training.

The final core Bicycle Principle (*Inclusion*) and the related accessibility and equity requirements must be actively prioritized since SFT's short duration and less formal context often leaves these features neglected. Inclusion is meant to be a blanket concept that, ideally, applies to all persons or groups. Inclusion means creating an environment where everyone feels welcome, valued, respected, and has equal opportunity for equivalent participation. In practice—particularly in the context of STEMM research and training—creation of inclusive environments can sometimes fail to consider the needs of all groups. For example, in situations where design is not co-developed, persons with disabilities are often left out of the "inclusion" conversation [54]. Therefore, it is necessary to treat inclusion more broadly and define accessibility as "an umbrella term for all aspects which influence a person's ability to function within an environment [55]." Treated as a core component of "inclusion", accessibility is the design and implementation of systems, policies, processes, ways of interacting, and environments to ensure that persons with disabilities have equivalent access to a given space, and therefore equivalent experiences when participating in an activity. These observations underpin Recommendation (D): *Operationalize Equitable and Inclusive Practice in Short-format Training as an Ethical Obligation*. Implementation of this recommendation could inform and support instructors with tools that help them support equity, inclusion, and accessibility and develop the mindset that these features are a minimum standard for professional practice. We also know that pervasive inequities and disparities in FHE continue to harm STEMM professionals even after they overcome barriers to advanced degrees [56]. Here, there is a positive opportunity to use SFT to correct disparity. Recommendation (E): *Deploy Short-format Training to Counter Inequity*, advocates for directing SFT resources to peoples who have been historically excluded from STEMM (e.g., minoritized ancestry groups, the disabled, low-income groups, the Global South). SFT resources should be thoughtfully and meaningfully deployed to counter inequity that may have resulted in historically excluded STEMM practitioners not receiving training opportunities that were available to others. In all these areas, actions are possible at all levels, and solutions must be co-created with the people they are intended to benefit.

A secondary reason why the Bicycle Principles and recommendations can succeed is that they work across the breadth of SFT, treating it as a system. If we consider the SFT "system" to be composed of its stakeholders, then we can impose some systematicity by developing stakeholder-focused solutions (e.g., Tables 2–5; see also Roadmap [57]) since stakeholder groups are one of the few features all SFT shares.

Confidence that stakeholders will use The Bicycle Principles and recommendations relies partly on their origin from within the community, partly from their alignment with ongoing SFT activities worldwide, and partly from the structure they offer to those who seek to improve SFT. These factors make change plausible. Reinholz et al. [30] concluded that two change theories are the most commonly used in FHE reform: Community of Practice [58] and Diffusion of Innovation [59]. Examples of SFT activities worldwide demonstrate that The Bicycle Principles and recommendations are supportive of both theories. For example, *The Carpentries* SFT

instructor training program [60] represents a community of practice consistent with The Principles and several recommendations (e.g., A, B, K, M). *The Carpentries* trainer curriculum requires instructors to be trained according to a set of evidence-based teaching standards, to integrate assessment into their two-day courses, and participate in discussion and feedback sessions to earn and maintain a credential. *ELIXIR-GOBLET* instructor training [61] and related *ELIXIR* training resources present examples of the Diffusion of Innovation theory, consistent with The Bicycle Principles and recommendations (e.g., A, B, I, N); innovative instructional tools such as the Bioinformatics Mastery Rubric [62] provide guidance for career-spanning learning, various workshops and professional forums are opportunities for instructors to be exposed to knowledge about a new method, persuaded by its benefits, and supported to implement, customize, and adopt. Using The Bicycle Principles as a framework to improve SFT creates the opportunity to learn from FHE reforms—making what could work within the structured FHE environment more transferable to SFT.

Change theories in FHE reforms differentiate between changes that come about from top-down policies or emerge from individual or group actions [31]. The Bicycle Principles orient all stakeholders to a common set of objectives, such that the recommendations can be partitioned into individual, collective, and policy-based actions (i.e., Tables 2–5). Several recommendations could result in policies or strategies that are prescribed top-down (e.g., recommendations: B, C, G, K, N), but many recommendations achieve their greatest impact through wide-spread adoption by individuals (e.g., A, E, H, I, L). We appreciate that the recommendations assume a level of autonomy and community engagement that might not be plausible for every potential implementer—most recommendations cannot be implemented by individual instructors alone. Just as in FHE reform, SFT instructors have responsibilities to enact some changes (e.g., Recommendation D), but success is unlikely if the burden of change rests exclusively with instructors [30]. Future work, including updates and customizations to a proposed Implementation Roadmap [57] will require creating a variety of approaches to bring recommendations into practice (e.g., checklists, instructor training, supportive infrastructures, policy mandates).

Finally, we note that although this work represents a consensus of experts, consensus cannot capture every possible circumstance. We leveraged the global reach of our organizing committee to recruit self-nominations, but limitations including COVID-19 meant we lacked direct representation from individuals in Africa, South and Central America, or Asia. However, we did have representation from organizations that have membership and activities in these regions. Starting in July 2022, the Bicycle Principles and recommendations have been widely disseminated online through bikeprinciples.org and through international conferences. Given the reach of the assembled group, and that online, in-person, and asynchronous dissemination, as well as focus groups have not surfaced any new or unaccommodated concerns, we believe the consensus derived by our group is likely to represent saturation on the topics. We do not take this to mean that more recommendations are not possible, only that we have arrived at a coherent set of recommendations. Within FHE STEMM education improvement efforts, a consistent finding is that success for such initiatives depends on considering the entire system in which the instruction occurs. Although Biswas et al. [33] and Reinholz et al. [30] are discussing FHE and undergraduate STEM(M) improvement, the SFT subject matter experts at our meeting identified recommendations for SFT-specific improvement that are similar to FHE-based guidance, albeit without the system-level structure of FHE. This post-hoc triangulation strengthens confidence in the validity of the results of this conference, while also highlighting the challenges facing individuals and communities in improving SFT.

To increase the likelihood that The Bicycle Principles and any recommendations are used, action is required at all stakeholder levels [30]. FHE reform efforts have engaged stakeholders

in several ways including funded research programs and institution-wide improvement projects. Journals and professional societies support dissemination of improvements. These mechanisms support SFT to a lesser extent; currently there is no comparable research program dedicated to SFT. SFT also lacks incentives that encourage innovations to be published, or that reward and recognize SFT instructors' accomplishments.

Despite fewer formal incentives, there is evidence that communities of practice could be a valuable mechanism for promoting adoption of The Bicycle Principles and recommendations. The "community" set of Bicycle Principles prompt SFT programs to think about how materials could be shared, and instructors recognized and incentivized. For example, *LifeSciTrainers* [63] is an informal online community of practice for individuals engaged in SFT in the life sciences. Through it, instructors meet monthly and use online forums to share ideas and materials independent of SFT instructors' affiliation with a specific program or topic area. Talk series highlight instructors' accomplishments and provide an opportunity to share innovations in an informal setting. *LifeSciTrainers* activities are consistent with advancing recommendations (A, C, G, H, I, J), and provide an example of approaches that could help share effective practices across programs. International participation in *LifeSciTrainers* suggests global enthusiasm for SFT communities.

Funders must also exercise their role in promoting SFT reform. Over time, and consistent with recommendation (F), grassroots efforts could provide the evidence that justifies funders in imposing top-down standards for the effectiveness and inclusiveness of the SFT they invest in. Recent successes can be emulated. The FAIR Principles [51] were proposed in 2016 to reform scientific data management, a highly complex and multidimensional topic (e.g., technology, policy, incentives). The FAIR Principles were widely adopted by stakeholders and were enshrined in institutional policies globally, including the U.S. National Institutes of Health (NIH) in 2023 [64].

Ultimately, the final and most important group to involve in SFT reform will be learners. Learners who are empowered to insist on quality would be a powerful force for change. Every learner should be able to expect effective and inclusive instruction. An important aim of The Bicycle Principles and recommendations is to transform SFT from a "black box"—a learning experience where learners are uncertain about efficacy and inclusion to "back of the box"—a learning experience where implementations of The Bicycle Principles serve as standardized and informative consumer "labels" which offer interpretable information on the efficacy, inclusion, and quality of instruction. Standardized, easy-to-compare SFT, would also benefit instructors and SFT funders.

## Conclusion

SFT improvement is urgent and achievable. As Deming and Noray concluded, "there is a widespread perception that STEM workers are in short supply. . . but it is the new STEM skills that are scarce, not the workers themselves [1]." The Bicycle Principles and associated recommendations organize what education research and the most effective SFT programs have learned, providing a rallying point for global SFT improvement efforts. SFT reform is a strategic long-term investment in the STEMM professionals we have spent decades developing, could accelerate the pace of discovery, and could broaden participation in STEMM. The rapid evolution of STEMM disciplines calls for optimizing SFT to make it more reliably effective, inclusive, and career-spanning.

STEMM practitioners need sustained and customized professional development to keep up with innovations. Short-format training (SFT) such as workshops and short-courses are relied upon widely but have unaddressed limitations. This project generated principles and

recommendations to make SFT consistently effective, inclusive, and career-spanning. Optimizing SFT could broaden participation in STEMM by preparing practitioners more equitably with transformative skills. Better SFT would also serve members of the STEMM workforce who have several decades of productivity ahead, but who may not benefit from education reforms that predominantly focus on undergraduate STEMM. The Bicycle Principles and accompanying recommendations apply to any SFT instruction and may be especially useful in rapidly evolving and multidisciplinary fields such as artificial intelligence, genomics, and precision medicine.

## Supporting information

**S1 Text. Short-format training (SFT) definition.** List of definitions used and agreed upon by the authors.
(DOCX)

**S2 Text. Community distribution.** Details on how call for participation was distributed.
(DOCX)

**S3 Text. Draft challenge vignette list for kick-off meeting.** Initial problem set considered by participants.
(DOCX)

**S4 Text. Precis.** Framing document used to orient participants to this project and its goals.
(DOCX)

**S5 Text. Complete example: Recommendation A.** Detailed example of a recommendation produced by the author group.
(DOCX)

**S6 Text. Full list of 14 recommendations.** A list of all recommendations developed and accepted by the authoring group.
(DOCX)

## Acknowledgments

The authors wish to thank Rebecca Leshan and the staff of the CSHL Banbury Center for their guidance and facilitation of our convenings, and other participants in the conference including Charla Lambert.

## Author Contributions

**Conceptualization:** Jason J. Williams, Rochelle E. Tractenberg.

**Data curation:** Jason J. Williams, Rochelle E. Tractenberg, Bérénice Batut, Erin A. Becker, Anne M. Brown, Melissa L. Burke, Ben Busby, Allissa A. Dillman, Samuel S. Donovan, Maria A. Doyle, Celia W. G. van Gelder, Christina R. Hall, Kate L. Hertweck, Kari L. Jordan, John R. Jungck, Ainsley R. Latour, Jessica M. Lindvall, Marta Lloret-Llinares, Gary S. McDowell, Rana Morris, Teresa Mourad, Amy Nisselle, Patricia Ordóñez, Lisanna Paladin, Patricia M. Palagi, Mahadeo A. Sukhai, Tracy K. Teal, Louise Woodley.

**Formal analysis:** Rochelle E. Tractenberg.

**Funding acquisition:** Jason J. Williams, Rochelle E. Tractenberg.

**Investigation:** Rochelle E. Tractenberg.

**Methodology:** Rochelle E. Tractenberg.

**Project administration:** Jason J. Williams.

**Resources:** Jason J. Williams.

**Validation:** Jason J. Williams, Rochelle E. Tractenberg, Bérénice Batut, Erin A. Becker, Anne M. Brown, Melissa L. Burke, Ben Busby, Allissa A. Dillman, Samuel S. Donovan, Maria A. Doyle, Celia W. G. van Gelder, Christina R. Hall, Kate L. Hertweck, Kari L. Jordan, John R. Jungck, Ainsley R. Latour, Jessica M. Lindvall, Marta Lloret-Llinares, Gary S. McDowell, Rana Morris, Teresa Mourad, Amy Nisselle, Patricia Ordóñez, Lisanna Paladin, Patricia M. Palagi, Mahadeo A. Sukhai, Tracy K. Teal, Louise Woodley.

**Visualization:** Jason J. Williams.

**Writing – original draft:** Jason J. Williams, Rochelle E. Tractenberg, Bérénice Batut, Erin A. Becker, Anne M. Brown, Melissa L. Burke, Ben Busby, Nisha K. Cooch, Allissa A. Dillman, Samuel S. Donovan, Maria A. Doyle, Celia W. G. van Gelder, Christina R. Hall, Kate L. Hertweck, Kari L. Jordan, John R. Jungck, Ainsley R. Latour, Jessica M. Lindvall, Marta Lloret-Llinares, Gary S. McDowell, Rana Morris, Teresa Mourad, Amy Nisselle, Patricia Ordóñez, Lisanna Paladin, Patricia M. Palagi, Mahadeo A. Sukhai, Tracy K. Teal, Louise Woodley.

**Writing – review & editing:** Jason J. Williams, Rochelle E. Tractenberg, Bérénice Batut, Erin A. Becker, Anne M. Brown, Melissa L. Burke, Ben Busby, Nisha K. Cooch, Allissa A. Dillman, Samuel S. Donovan, Maria A. Doyle, Celia W. G. van Gelder, Christina R. Hall, Kate L. Hertweck, Kari L. Jordan, John R. Jungck, Ainsley R. Latour, Jessica M. Lindvall, Marta Lloret-Llinares, Gary S. McDowell, Rana Morris, Teresa Mourad, Amy Nisselle, Patricia Ordóñez, Lisanna Paladin, Patricia M. Palagi, Mahadeo A. Sukhai, Tracy K. Teal, Louise Woodley.

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
