## [Decision Letter · Decision Letter 0]

4 Sep 2023

PONE-D-23-21707An International Consensus on Effective, Inclusive, and Career-spanning Short-format Training in the Life Sciences and BeyondPLOS ONE

Dear Dr. Tractenberg,

Thank you for submitting your manuscript to PLOS ONE. After careful consideration, we feel that it has merit but does not fully meet PLOS ONE’s publication criteria as it currently stands. Therefore, we invite you to submit a revised version of the manuscript that addresses the points raised during the review process.

Notes from editor: Thank you for your submission to Plos One. The reviewers have suggested some minor changes to the paper before it is ready for publication (outlined below). Please pay particular attention to comments about data availability. Please also amend the abstract to briefly mention the delphi method and the participant demographics.  Best of luck with the resubmission.

We look forward to receiving your revised manuscript.

Kind regards,

Cathryn Knight

Academic Editor

PLOS ONE

Journal Requirements:

   "This material is based upon work supported by the National Science Foundation under DRL/EHR:2027025. Any opinions, findings, and conclusions or recommendations expressed in this material are those of the author(s) and do not necessarily reflect the views of the National Science Foundation. The authors wish to thank Rebecca Leshan and the staff of the CSHL Banbury Center for their guidance and facilitation of our convenings, and other participants in the conference including Charla Lambert."

  "This work was supported by the National Science Foundation under DRL/EHR: 2027025 to JWW & RET. Funders played no role in the study design, data collection and analysis, decision to publish, or preparation of the manuscript."

   "B.B.: I have read the journal's policy and  have the following competing interest: I am an employee of DNAnexus.

G.S.M.: I have read the journal's policy and have the following competing interest: I work as an independent consultant who provides short-format training and professional development services. 

K.L.J.: I have read the journal's policy and have the following competing interest: I am an employee of The Carpentries.

E.A.B.: I have read the journal's policy and have the following competing interest: I am an employee of The Carpentries. 

N.C.: I have read the journal's policy and have the following competing interest: I am the owner of CureComms Advisors LLC and was compensated for participating in the meeting and providing strategic content guidance. 

No other authors declare competing interests."   

We note that one or more of the authors are employed by a commercial company: DNAnexus and The Carpentries.

4. We noted in your submission details that a portion of your manuscript may have been presented or published elsewhere. [The manuscript was submitted to biorxiv in march 2023. The submitted version is a more concise version (same results), and is formatted for PLOS one. Williams J, Tractenberg RE, Batut B, Becker E, Brown A, Burke M, et al. (2023, 15 March). Optimizing Short-format Training: an International Consensus on Effective, Inclusive, and Career-spanning Professional Development in the Life Sciences and Beyond. bioRxiv 2023.03.10.531570; doi: https://doi.org/10.1101/2023.03.10.531570] Please clarify whether this [conference proceeding or publication] was peer-reviewed and formally published. If this work was previously peer-reviewed and published, in the cover letter please provide the reason that this work does not constitute dual publication and should be included in the current manuscript.

Additional Editor Comments:

-mentioned above

Reviewers' comments:

Reviewer's Responses to Questions

**Comments to the Author**

1. Is the manuscript technically sound, and do the data support the conclusions?

Reviewer #1: Yes

Reviewer #2: Yes

2. Has the statistical analysis been performed appropriately and rigorously? 

Reviewer #1: I Don't Know

Reviewer #2: N/A

3. Have the authors made all data underlying the findings in their manuscript fully available?

Reviewer #1: No

Reviewer #2: No

4. Is the manuscript presented in an intelligible fashion and written in standard English?

Reviewer #1: Yes

Reviewer #2: Yes

5. Review Comments to the Author

Reviewer #1: About “statistical analysis”: I selected “I don’t know” because the authors use content analysis, but don’t discuss much about what they did.

About data availability: I selected “No”, though what they have done might be acceptable. The authors don’t provide the (de-identified) documents on which they did content analysis, at least as far as I can discern. It would not be possible to reproduce their work. Perhaps these “raw data” are not required to be shared. Defer to editor.

Other comments:

This is an interesting manuscript from which emerges worthwhile messages for improving learning throughout people’s scientific careers. I’m sure the discussions that went into the development of these principles and recommendations for short-format training were fascinating.

Lines 145-164. This is a lot of text to explain essentially what is in the figure. Comes across as repetitive.

Line 172-3: Subtle difference between the text and figure here. The figure is, I think, more accurate, by stating "formal knowledge about teaching and learning applies". The authors note in more than one place that the instructors of SFT often don't have pedagogical training (i.e. "formal knowledge about teaching and learning", and as an aside, neither do many, many instructors in FHE). So this formal knowledge is not necessarily a shared characteristic in the real world. But it ought to "apply" to both.

Suggestion: perhaps reword slightly to "...the fact that formal knowledge about teaching and learning ought to apply to both forms is perhaps the only shared characteristic"

Line 197-99 I think this sentence might be missing a word. "...adopt them"?

Line 234 (and lots of other places) The 20 items listed in S3 look like a list of questions for discussion – this is not my understanding of the word "vignette". I do recognize that the authors are reporting on meetings they had where they used “vignette”, but it seems like an odd choice.

Line 285-86. The context stated here that most of the participants are doing this work in countries outside where they are based seems important. Are these folks primarily involved in development work in developing countries (e.g. providing SFTs in topics that are not available locally)? If so, are these settings in which the participants work a relevant aspect of the principles generated? Does this bound the applicability of the principles? Are they particularly suited for this context?

Figure 2. The rear wheel is shown as a "cycle", but it is not clearly explained why these four parts connect in a cycle. In fact, the image shows the four items in a different order than the bullet points below the image. I'd recommend the authors figure out how to better connect the parts of the cycle, if it is one. As a reader, I think these four are more like a good list, though a case could be made for a cycle (e.g. what links "Inclusive" back to "Best Evidence"? Possibly collecting evidence about the inclusivity of a learning experience will contribute to "best evidence"...). The justification for a "cycle" is not communicated yet, in my opinion.

For the front wheel, I think the authors are implying an amplifying feedback loop, where if you reach more learners, you need to figure out how to offer more courses or make the courses bigger, or involve more instructors, and, if you want to continue to grow (reach more learners), you need to ensure you have systems in place to support that growth. Reaching more people with effective and inclusive SFTs, and making this possible by sharing good materials and coordinating systems are excellent goals. This wheel is easier to see as a cycle, but what's the end point of this amplifying feedback loop? Saturation of the learner pool?

In general, although I love bicycles, this image seems a bit forced.

Line 335: I believe this should reference "S5" instead of "S4.

S6 and Tables 2-5: I recommend that in S6 and in the tables that the authors number the 14 items, rather than use letters. The letters are confusing. E.g. in Table 2, the four recommendations spell FILM. This is a totally unnecessary distraction. Just number these 1-4, then start with 5 in the next grouping. I'm guessing there was much discussion about not wanting to number things which might imply a priority, but lettering simply muddles your list for a reader.

If you don't want to lead with what is currently Recommendation F, lead with L – L (catalysis), to me, is at the heart of this entire effort.

Table 2: FAIR is used here as an acronym, but these principles are not referenced until line 532.

Lines 352-353. When you say "individuals", do you mean individuals working essentially alone? Or do you mean individual instructors who are part of a larger group of SFT providers who collect evidence from their courses? This sentence implies that funders are most likely to listen to individuals working alone, which might not be what you mean. For example, you note below that it may be difficult for individuals to convince their colleagues to share goals and standards. Funders may be a step harder for individuals.

Reviewer #2: The manuscript by Williams et al. provides a report on the outputs of a project which aimed to improve the delivery and impact of Short Format Training (SFT) workshops. The manuscript describes the process that the project went through to develop the final recommendations that were produced, and then explores the potential implementation of these recommendations across different "user groups" (i.e., individual trainers, communities, institutions).

The manuscript is well-written, and does a good job of describing the current state of SFT-based training initiatives (including coverage of current challenges), before moving on to explain the processes that the authors went through to generate the proposed stet of recommendations.

Of note, the authors' highlighting of the relatively low-impact of current SFT (i.e., in terms of how much long-term utility attendees actually gain from this type of training) is extremely valuable (although also rather sobering), particularly in the light of how much has been invested in developing and delivering this style of training in recent years. Rather than dismissing this approach to content delivery, the authors are proposing some useful and readily-implementable recommendations to help improve SFT impact.

Given the popularity and ubiquity of Short Format Training across a wide range of disciplines, this thoughtful and well-structured manuscript is a timely and highly relevant submission.

Specific comments:

1. Line 235 - given the breadth of the potential target audience (i.e., almost anyone involved in developing, delivering and/or coordinating SFT) it might be worth briefly defining "phenomenography", which will not necessarily be something that non-qualitative researchers will be familiar with.

2. Lines 289-290: it would be useful to see the numbers involved from each country (e.g., "Canada (x), France (y)" etc).

3. Lines 296-298: Presumably the "average" used here was the mean? (this should be specified). Given the highly skewed nature of these figures (e.g., 0-7000 for "new instructors trained"), the median would be a more appropriate measure of the "average". Also, based on the journal's requirements for data availability, the raw data behind these averages needs to be made available (this could easily be done as a small spreadsheet in the supplementary information).

4. Line 302: "Catalytic learning" should be defined somewhere in the document. I realise that it is loosely defined in Figure 2, but I think a more foraml defintion in the text (or within a specific box - see next comment), along with a relevant citation, would be helpful.

5. Line 320: as per my comment above about "phenomenography", I think it would be worth briefly defining the Delphi technique here. I wonder if having a "Definitions Box" might be useful, just to help readers who are not coming from an educational research background?

6. Line 342: "i.e.," should be written as "that is" if it appears in the main text (rather than within parentheses).

7. Line 363: it isn't obvious what is meant by "professionalizing" in this context - this should be made clear. It could mean "utilsing a formal paid training system" (which I'm assuming is the intent), but it could also be interpreted as "acting in a more professional manner".

8. Line 367: the term "formal organizations" is used here without definition. The examples associated with "organizations and institutions" in the next paragraph almost look like a definition for "formal organizations" - could this definition be moved up so that it is clear what "formal organizations" is referring to?

9. Lines 431-432: reference 50 is talking specifically about ableism, however the authors' mention of "creation of inclusive environments" would generally be taken to refer to more than persons with disabilities (e.g., minorities, neurodiverse individuals, indigenous communities etc). I think the wording in this part of the paragraph needs to be modified to make sure that "inclusive environments" is an broad as possible, and then rephrase the ableist-specific material as one example of how to co-design for inclusivity.

6. PLOS authors have the option to publish the peer review history of their article (what does this mean?). If published, this will include your full peer review and any attached files.

Reviewer #1: No

Reviewer #2: No

---

## [Author Response · Author response to Decision Letter 0]

13 Oct 2023

Responses to reviewers are outlined in the attached document.

---

## [Editor Report · Decision Letter 1]

23 Oct 2023

An International Consensus on Effective, Inclusive, and Career-spanning Short-format Training in the Life Sciences and Beyond

PONE-D-23-21707R1

Dear Dr. Tractenberg,

We’re pleased to inform you that your manuscript has been judged scientifically suitable for publication and will be formally accepted for publication once it meets all outstanding technical requirements.

Kind regards,

Cathryn Knight

Academic Editor

PLOS ONE

---

## [Editor Report · Acceptance letter]

31 Oct 2023

PONE-D-23-21707R1 

An International Consensus on Effective, Inclusive, and Career-spanning Short-format Training in the Life Sciences and Beyond 

Dear Dr. Tractenberg:

I'm pleased to inform you that your manuscript has been deemed suitable for publication in PLOS ONE. Congratulations! Your manuscript is now with our production department. 

Kind regards, 

on behalf of

Dr. Cathryn Knight 

Academic Editor

PLOS ONE